# Cancer Cytotoxicity of a Hybrid Hyaluronan-Superparamagnetic Iron Oxide Nanoparticle Material: An In-Vitro Evaluation

**DOI:** 10.3390/nano12030496

**Published:** 2022-01-31

**Authors:** Yen-Lan Chang, Pei-Bang Liao, Ping-Han Wu, Wei-Jen Chang, Sheng-Yang Lee, Haw-Ming Huang

**Affiliations:** 1Divison of Prosthodontics, Department of Stomatology, Mackay Memorial Hospital, Taipei 10449, Taiwan; fabricate-duck@hotmail.com; 2Department of Dentistry, Taipei Medical University Hospital, Taipei 11031, Taiwan; 927004@h.tmu.edu.tw; 3Graduate Institute of Biomedical Materials and Tissue Engineering, College of Biomedical Engineering, Taipei Medical University, Taipei 11031, Taiwan; gahwclbjwph@hotmail.com; 4School of Dentistry, Taipei Medical University, Taipei 11031, Taiwan; m8404006@tmu.edu.tw (W.-J.C.); seanlee@tmu.edu.tw (S.-Y.L.); 5Department of Dentistry, Wan-Fang Medical Center, Taipei Medical University, 11696 Taipei, Taiwan

**Keywords:** hyaluronan, superparamagnetic iron oxide nanoparticle, cancer, TOF-SIM

## Abstract

While hyaluronic acid encapsulating superparamagnetic iron oxide nanoparticles have been reported to exhibit selective cytotoxicity toward cancer cells, it is unclear whether low-molecular-weight hyaluronic acid-conjugated superparamagnetic iron oxide nanoparticles also display such cytotoxicity. In this study, high-molecular-weight hyaluronic acid was irradiated with γ-ray, while Fe_3_O_4_ nanoparticles were fabricated using chemical co-precipitation. The low-molecular-weight hyaluronic acid and Fe_3_O_4_ nanoparticles were then combined according to a previous study. Size distribution, zeta potential, and the binding between hyaluronic acid and iron oxide nanoparticles were examined using dynamic light scattering and a nuclear magnetic resonance spectroscopy. The ability of the fabricated low-molecular-weight hyaluronic acid conjugated superparamagnetic iron oxide nanoparticles to target cancer cells was examined using time-of-flight secondary ion mass spectrometry and T2* weighted magnetic resonance images to compare iron signals in U87MG human glioblastoma and NIH3T3 normal fibroblast cell lines. Comparison showed that the present material could target U87MG cells at a higher rate than NIH3T3 control cells, with a viability inhibition rate of 34% observed at day two and no cytotoxicity observed in NIH3T3 normal fibroblasts during the three-day experimental period. Supported by mass spectrometry images confirming that the nanoparticles accumulated on the surface of cancer cells, the fabricated materials can reasonably be suggested as a candidate for both magnetic resonance imaging applications and as an injectable anticancer agent.

## 1. Introduction

Hyaluronic acid (HA) is a biopolymer based on natural polysaccharides. It is a linear non-sulfated glycosaminoglycan found in abundance in connective, epidermal, neural, and joint tissues, and is the main component of the extracellular matrix in the human body [1,2,3]. HA is wildly used in medical treatment of osteoarthritis, skin wounds, and dry eye disease. In addition, due to its biocompatibility and non-toxicity properties, HA is frequently used as a biomaterial for controlled drug release [2,3]. Notably, HA has been found to bind to the receptor of cluster of differentiation maker-44 (CD44), which is found in abundance on cancer cell membranes [4,5], and because tumor selectivity is an important issue for developing cancer treatment agents. Interest has increased in HA’s possible use as a tumor-targeting agent [6,7], both for the development of superior natural materials for MR imaging and as an anti-tumor drug delivery system.

HA demonstrates many benefits for controlled drug release, such as accurately maintaining drug concentrations and efficiently carrying hydrophobic drugs while having low toxicity to normal tissue. It has been reported that the commonly used chemotherapy drugs, including paclitaxel and doxorubicin have shown superior experimental results when HA was used as a carrier [2]. Inorganic nanoparticle (NPs), including iron oxide, gold, and silica, have been examined in the literature on imaging [8,9]. However, due to NPs’ potential cytotoxicity and low cell targeting ability, HA is often used to improve the material’s properties; in several experimental studies, for example, superparamagnetic iron oxide NPs (SPIONs) have been used to fabricate an MRI contrast agent [10,11]. HA can also be used as an interface for the fabrication of hybrid image enhancing materials. For example, Lee et al. used near-infrared fluorescence dye-labeled HA as a coating for SPIONs to create an image enhancer for both magnetic resonance and enzyme-sensitive optical imaging [12].

Interestingly, recent studies have demonstrated that HA-SPIONs can also function as therapeutic agents [3]. When HA conjugates on NP surfaces, selectivity towards cancer cells can be enhanced through active targeting [1,2,7]. Several examples of such nanomaterials that enhance the overall pharmacological properties of anticancer drugs have been reported [1,2], such as a report by Fu et al., indicating that surface functionalization of SPION by HA exhibits good water dispersibility and enhances magnetic relaxivity for MR imaging. In addition, such a hybrid material significantly increases cellular uptake and specific accumulation in human hepatocellular liver carcinoma HepG2 cells, providing a reference for effective chemotherapy for hepatocellular carcinoma [13,14]. El-dakdouki and coworkers’ in vitro and in vivo experiments found that HA-SPIONs enhanced the efficacy of the conjugated anticancer drug doxorubicin in treating ovarian cancer [15]. Smejkalova et al. (2014) also found that HA-SPION itself exhibited anticancer effects toward various cancer cell lines, including human colorectal carcinoma (HT-29, HCT-116 and Caco-2), hepatocellular carcinoma (C3A) as well as lung (A549) and breast adenocarcinoma (MCF7 and MDAMB231). Most importantly, HA-SPION has shown a high level of selectivity towards cancer cells while having no toxic effects on normal fibroblasts [16]. This finding provides an insight into developing new anti-cancer metrics.

Long chain HA can be depolymerized into low molecular weight hyaluronic acid (LMWHA) by various physical, chemical, and biological methods [17]. After HA is broken down, its physical and physiological function is strongly correlated to molecular weight, which varies in natural HA from several kDa (HA oligomers) to larger than 5 MDa (high-molecular weight HA, HMWHA) [18,19,20]. HA with different molecular weight can lead to different, and sometimes opposing, biological functions. For example, HMWHA has been reported to be both anti-inflammatory and immunosuppressive, LMWH As are associated with inflammatory stimulation and angiogenesis, and medium-size HA promotes wound healing and regeneration [3,21,22]. Kuo et al., (2021) investigated the acceleration of new bone formation using a high and low molecular weight hyaluronic acid hybrid and found that defects filled with LMWHA exhibited more new bone formation compared to those filled with HMWHA [23].

Because HA can bind with CD44 as mentioned above, fabricating HA-SPION hybrids to enhance the ability of superparamagnetic nanoparticles to target cancerous tissue has attracted increasing attention. As with inflammation and immunosuppression, the ability of HA-SPION hybrids to target cancer is also affected by molecular weight. Zhong et al. (2019) used HA at molecular weights of 7, 63 and 102 kDa to fabricate HA-SPIONs for drug release and found CD44 receptor targeting to vary according to molecular weight [24]. As of yet, however, no reports discuss how reducing molecular weight might affect the HA-SPIONs’ selectivity and anti-cancer efficiency. Accordingly, the present study seeks to evaluate LMWHA-SPION’s ability to inhibit cancer cell viability.

## 2. Materials and Methods

### 2.1. Materials

FeCl_2_•4H_2_O and Hyaluronic acid (molecular weight 3000 kDa) were obtained from Avantor Performance Material, Inc. (Radnor, PA, USA) and Cheng-Yi Chemical Industry Co. Ltd. (Taipei, Taiwan), respectively. FeCl_3_•6H_2_O, oleic acid and sodium nitrate were purchased from Nacalai Tesque (Kyoto, Japan). Dimethyl sulfoxide (DMSO), 3-(4,5-Dimethylthiazol-2-yl)-2,5-diphenyltetrazolium bromide (MTT), and agarose were obtained from Sigma-Aldrich (St. Louis, MO, USA). Dulbecco’s modified Eagle medium (DMEM), fetal bovine serum (FBS), and penicillin-streptomycin were purchased from Gibco (Grand Island, NY, USA). All other solvents and reagents were purchased from J.T. Baker (Phillipsburg, NJ, USA).

### 2.2. Preparation of Oleic-Acid Coated Fe_3_O_4_ Nanoparticles

Fe_3_O_4_ nanoparticles (SPIONs) were prepared using co-precipitation. As presented in a previous study [25], FeCl_2_•4H_2_O and FeCl_3_•6H_2_O were mixed in distilled water then mixed with NH_4_OH to create an alkaline solution. The finial material ratio for FeCl_2_•4H_2_O, FeCl_3_•6H_2_O, and NH_4_OH was 1:2.5:2.5. Fe_3_O_4_ NPs formed after cooling the solution to 25 °C (Figure 1a). The final chemical reaction is listed as follows:Fe^2^^+^ + 2Fe^3^^+^ + 8OH^-^ → Fe_3_O_4_ + 4H_2_O

The nanoparticles were coated with fatty acid by adding oleic acid at 85 °C followed by 30 min stirring. Oleic-coated SPIONs were collected using a neodymium magnet and re-dissolved with purified water after washing (Figure 1a). The prepared oleic-coated SPIONs were observed under a transmission electron microscope (TEM, H-600, Hitachi, Ltd., Tokyo, Japan). Particle size of oleic-coated SPIONs were detected using an electrophoretic light scattering device (NanoBrook 90Plus Zeta, Brookhaven Instruments, Holtsville, NY, USA). To confirm the fabricated oleic-coated SPIONs’ superparamagnetic properties, the hysteresis loops of prepared nanoparticles were detected using a superconducting quantum interference device (SQUID) (MPMS7, Quantum Design, San Diego, CA, USA) at temperatures of 5 K and 300 K.

### 2.3. Manufacture of LMWHA-SPIONs

The LMWHA used in this study was produced by irradiating the purchased HMWHA with a cobalt-60 irradiator (Point Source, AECL, IR-79, Nordion, Ottawa, ON, Canada), as described in a previous study [26]. The irradiating dose was set at 1 kGy/h for 20 h continuous exposure. Gel permeation chromatography (GPC) was performed to measure the molecular weight of the produced LMWHA. Before detection, the degraded HA was dissolved in 0.1 M NaCl. Then, the mixed solution was analyzed using a separation device (Series 200, Perkin Elmer, Waltham, MA, USA) connected with a size-exclusion column (SB-806M HQ, Shodex, Kanagawa, Japan) in a 25 °C incubator. Sodium nitrate (0.1 M) was used as a mobile phase and separated samples were collected using a refractive index detector (Series 200, Perkin Elmer, Waltham, MA, USA). A standard curve was obtained using a standard particle kit at various molecular weights (Pullulan ReadyCal Kits, PSS Polymer Standards Service, Mainz, Germany). The molecular weight of the prepared LMWHA sample was obtained using commercial software (ChromManager 5.8, ABDC WorkShop, Taichung, Taiwan).

LMWHA-SPIONs were synthesized according to a previous study [16]. Successful synthesis was confirmed by the signal of chemical shift at 0.9 ppm of the 1H nuclear magnetic resonance (NMR) spectra (DRX500 Avance, Bruker BioSpin GmbH, Rheinstetten, Germany). Particle size of the fabricated neat LMWHA and LMWHA-SPIONs were measured as above for SPIONs.

### 2.4. Targeting Ability of LMWHA-SPIONs

In this study, normal fibroblast (NIH3T3) and glioblastoma (U87MG) cell lines were used. Cells were seeded with Dulbecco’s modified eagle medium (DMEM) supplemented with 10% fetal bovine serum (FBS) and 1% penicillin-streptomycin at a density of 1 × 10^4^ cells/mL and incubated at 37 °C with a 5% CO_2_ supplement. To test the selective targeting ability of prepared LMWHA-SPIONs on the two cultured cells, time-of-flight secondary ion mass spectrometry (TOF-SIMS) and T2*-weighted magnetic resonance images were obtained. During TOF-SIMS analysis, both NIH3T3 and U87MG cells with a density of 5 × 10^5^ cells/mL were cultured in 6-well plates. After 24 h, the prepared LMWHA-SPIONs at a concentration of 1 mg/mL were added, followed by an additional 24 h culture. Then, the cells were washed and fixed with glutaraldehyde. A Bi^3+^ primary ion beam (30 keV) was applied to the sample surface (200 × 200 µm^2^) and emitted ions collected using time-of-flight secondary ion mass spectrometry (TOF-SIMS) (PHI TRIFT IV, ULVAC-PHI, Kanagawa, Japan). In this study, *m*/*z* 56 (Fe ion), *m*/*z* 86 and *m*/*z* 184 (phospholipid) signals were detected and used to build ion distribution images.

For T2*-weighted magnetic resonance imaging, phantom samples of NIH3T3 and U87MG cells cultured with of 2 mg/mL LMWHA-SPIONs were prepared as per a previous report [19]. LMWHA-SPIONs labeled cells with a concentration of 1.0 × 10^5^ cells/mL were added to agarose gel. After using ultrasound to remove air bubbles, cells were covered by an additional layer of agarose. A 1.5 T MRI device (Signa HDxt superconductor clinical MR system, GE Medical Systems, Milwaukee, WI, USA) was used to obtain MRI images of the LMWHA-SPIONs targeted cells. After placing the phantom dish into the MRI chamber, T2* weighted images were acquired with the machine set to the following parameters: 170 ms repetition time, 7 ms echo time, echo number of 1. Image size was set to 256 × 256 with pixel bandwidth of 115. Acquired images were quantified by measuring the gray intensity of gel samples using scientific analysis software (Image Pro Plus, Media Cybernetics, Inc., Silver Spring, MD, USA).

### 2.5. Viability of LMWHA-SPIONs Targeted Cells

Viability of NIH3T3 and U87MG cells was tested both with and without the prepared LMWHA-SPIONs. After 24 h culture for cell attachment, the supernatant was removed and a new medium containing 2 mg/mL LMWHA-SPIONs was added to the cultured plate. Fresh medium without LMWHA-SPIONs was set as the control. Cell viability was determined using the MTT (3-(4,5-Dimethylthiazol-2-yl)-2,5-diphenyltetrazolium bromide) method. Absorbance at 570/690 nM was read every 24 h for three days using a microplate reader (EZ Read 400, Biochrom, Holliston, MA, USA). Statistical differences between test and control samples were obtained through one-way analysis of variance (ANOVA) with Tukey’s post hoc tests (SPSS Inc., Chicago, IL, USA). In this study, statistical significance was set at a *p*-value lower than 0.05.

## 3. Results and Discussion

Surface modification is a common technique to prevent SPION oxidation and the release of possibly cytotoxic metal ions [16,27]. Surface modification can also improve SPION hydrophilicity. To avoid particle aggregation, the SPIONs prepared in this study were coated with oleic acid to create a high level of colloid stability, as proposed by several reports [16,28,29]. TEM images (Figure 1c), of the oleic acid coated SPIONs showed minor aggregation in water and particle sizes distributed mainly (79%) in the range of 4 to 10 nM (Figure 1d). Since particle sizes were under 30 nM, these prepared SPIONs could pass through a superparamagnetic-ferromagnetic transition [30,31]. As shown in Figure 1e, the hysteresis loop of prepared OA-coated SPIONs could be found at 5 K, but not at 300 K, confirming the superparamagnetic behavior of these SPIONs and their usefulness in MR imaging applications [32].

In this study, LMWHA was prepared through γ-irradiation exposure at a dose of 20 kGy. The final molecular weight of prepared LMWHA was about 250 kDa. Figure 2a,b, respectively, show the NMR spectrums of neat LMWHA and LMWHA-SPIONs prepared in this study. Successful binding between oleic acid and LMWHA can be confirmed through the existence of oleic acid long chain proton signals at 1.6 ppm [16]. Figure 2c and d showed the particle size distributions of the neat LMWHA and LMWHA-SPIONs, respectively. The particle sizes of modified SPIONs increased slightly by changing the main distribution from 400–700 nm to 600–800 nm. The zeta potential and polydispersity of these prepared LMWHA-SPIONs were measured at −43.84 mV and 0.37, respectively; values in line with previous reports [15,18,33]. The high negative charge is contributed to by the carboxylate base of D-glucuronic acid in the HA backbone [16], which provides strong electrostatic repulsion and results in prevention of aggregation in an aqueous solution [18]. The major concern for the use of HA as an injectable drug-loading agent is its high viscosity, which may cause the substance to aggregate in blood vessels [16]. Since reducing HA’s molecular weight can sharply decrease its viscosity; these fabricated LMWHA-SPIONs appear to be a feasible tool for the development of injectable anti-cancer agents [17].

Because of its sensitivity in detecting cells containing inorganic substances [34,35] and its proven ability to detect specific binding between cells and HA-IONPs [17,18], TOF-SIMS was used to confirm specific binding of LMWHA-SPIONs to U87MG cancer cells in this study. During TOF-SIM analysis, a high-energy ion beam was used to ionize sample molecules before ionized molecular fragments (known as secondary ions) were captured using a time-of-flight mass spectrometry. The mass-to-charge ratios (*m*/*z*) of collected secondary ions were recorded according to their emitted surface [36]. For cellular analysis, the two major indicators for identifying the cellular membrane are C_5_H_12_N^+^ fragments (*m*/*z* 86) and the phosphatidylcholine head group (C_5_H_15_NPO_4_^+^) (*m*/*z* 186) [17,18,37,38,39]. When NIH3T3 and U87MG cells were co-cultured with prepared LMWHA-SPIONs, a differential ability to target the two cell types was observed (Figure 3). Fe ions signals (*m*/*z* 56) were higher in U87MG cells compared to NIH3T3 samples (Figure 4).

T2* weighted MR images were also used to confirm the differential ability of LMWHA-SPIONs to bind to U87MG cancer cells and NIH3T3 normal fibroblast, which occurs because superparamagnetic materials such as magnetite with a diameter less than 30 nM exhibit excellent superparamagnetic properties. Thus, the LMWHA-SPIONs prepared in this study can be used as a T2 contrast agent to alter T2 relaxation time and relaxivity of caner tissue [13,14,40]. For both cell lines, the intensity of T2* weighted MR images was reduced in samples co-cultured with LMWHA-SPIONs (Figure 5). Comparing images Figure 5c,d, the high contrast performance of prepared polymer-coated Fe_3_O_4_ NPs replicated a previous study [41]. However, the signal intensity of U87MG cells was 1.24 times higher than NIH3T3 cells when the cells were treated with LMWHA-SPIONs (Figure 5e). From results shown in Figure 3 and Figure 5, we could confirm that LMWHA-SPIONs provided an effective means of distinguishing tumor cells from normal ones. The selective binding of LMWHA-SPIONs to U87MG cancer cells and NIH3T3 normal fibroblasts was consistent with a previous study [18], and suggested the potential for the material’s use in cancer detection and treatment.

Two major concerns for developing a cancer treatment agent are reducing toxicity and enhancing tumor selectivity. Smejkalova et al., (2014) showed, for the first time, that, unlike how HA-SPION can increase the proliferation of normal fibroblast cells, HA-SPION is also cytotoxic toward a number of human cancer cell lines, including human colorectal carcinoma (HT-29, HCT-116 and Caco-2) and hepatocellular carcinoma (C3A), as well as lung (A549) and breast adenocarcinoma (MCF7 and MDAMB231) [16]. In this study, HA-SPION was found to be cytotoxic based on polymeric micelles that can also be found in human glioblastoma. When cancer cells were cultured with the prepared LMWHA-SPIONs, a significant reduction in viability was found over the course of three days (Figure 6a). However, this toxicity was not detected in normal fibroblast cells (Figure 6b). The inhibition rate for U87MG cells was 75%, 66% and 70% over the three-day experimental period (Figure 6c), confirming the anticancer effect of LMWHA-SPIONs, as per research by Smejkalova et al. [16]. It has been reported that the HA’s cellular binding ability depends strongly on its molecular weight [42]. Since U87MG and NIH3T3 cells were both co-cultured with the same polymeric micelles, the effect of molecular weight on the selective cytotoxicity demonstrated in this study can be ruled out. As mentioned above, HA has also shown a high specific binding ability to cancer cells via CD44 on the cellular membrane [1,2,5,42]. Since a higher number of CD44 receptors are found on the cellular membranes of glioblastoma cells than on normal fibroblast cells [43,44,45], the targeting (Figure 3 and Figure 5) and cytotoxicity (Figure 6) of the prepared LMWHA-SPIONs was greater for U87MG glioblastoma than NIH3T3 normal fibroblasts.

However, the number of CD44 receptors on cell surfaces seems to be only one mechanism related to HA-SPIONs’ anticancer effect. As in Smejkalova et al.’s report, HA-SPION was most cytotoxic to HT-29 compared to other cancer cell lines. In contrast, human malignant melanoma (A-2058) treated with HA-SPIONs showed the lowest inhibition in cell viability. However, the expression of CD44 is comparable in both HT-29 and A-2058 cells. The authors found that the amount of intracellular iron post HA-SPION cell treatment was also a factor [16]. Their in vitro experiment and in vivo histological examinations indicated that aggregated iron particles were located on the surface of tumor cells, but nonaggregated solubilized iron particles were detected inside the normal cells. Since the intracellular concentration of iron ions has been reported as a factor that increases DNA synthesis essential for cell proliferation [46]; this may explain why HA has an opposite effect in cancer and normal cells. In Figure 7, *m*/*z* 56 and 86 convolution images of U87MG cells cultured with LMWHA-SPIONs also indicate the presence of Fe_3_O_4_ NPs on the cellular surface. This result is both consistent with previous research [18] and provides visual evidence supporting the findings by Smejkalova et al. [16].

## 4. Conclusions

Low molecular weight hyaluronan-oleic acid-coated SPIONs were fabricated in this study. The successful binding between oleic acid and hyaluronic acid was confirmed through the existence of oleic acid long chain proton signals of NMR spectrum. Specifically bind to U87MG glioblastoma cells was proven by TOF-SIM- and T2*-weighted MR images, demonstrating application as an MR contrast agent. Cytotoxicity analysis showed that this polymeric micelle demonstrated an anticancer effect on glioblastoma cells while being harmless to normal fibroblasts. This selective cytotoxic behavior in vitro suggests that fabricated LMWHA-SPIONs could be a valuable anticancer agent. Taken together, the results of this study indicate a potential dual use of LMWHA-SPION micelles for precision therapy and diagnosis of tumor tissue.

## Figures and Tables

**Figure 1 nanomaterials-12-00496-f001:**
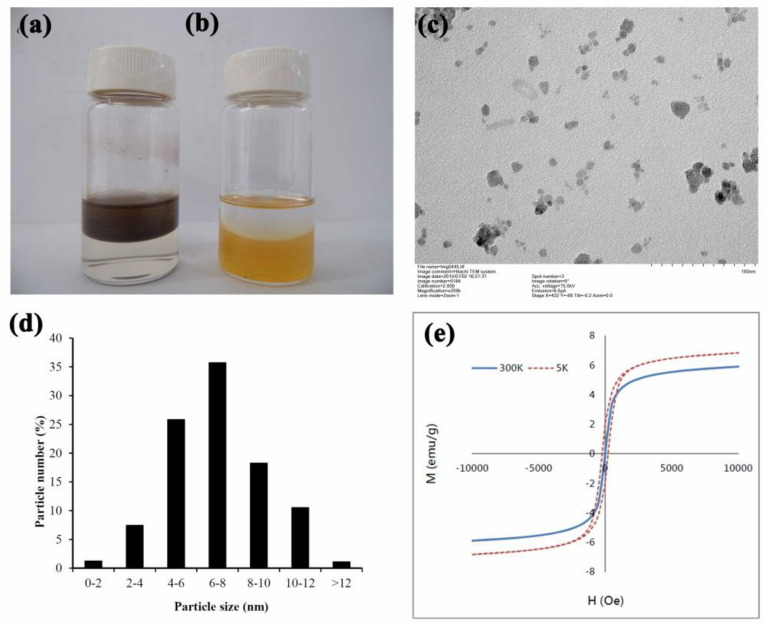
Prepared oleic acid-coated SPIONs (**a**) and LMWHA-SPIONs (**b**). TEM image of prepared SPIONs (**c**). The particle size distribution of the fabricated SPIONs mainly concentrated in the range of 4 to 10 nM (**d**). The absence of hysteresis loop confirms the superparamagnetic behavior of the prepared OA-coated SPIONs at 300 K (**e**).

**Figure 2 nanomaterials-12-00496-f002:**
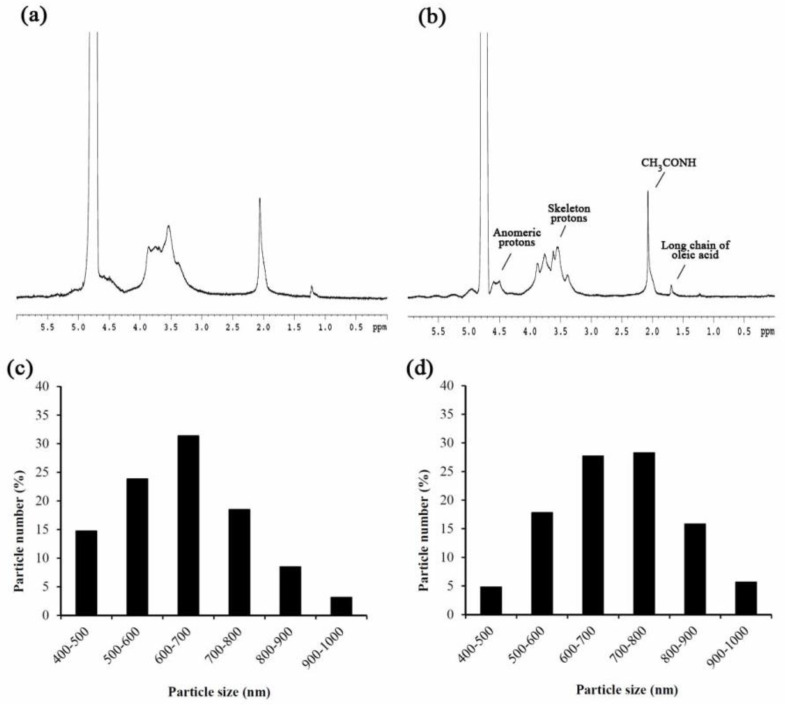
Successful binding of SPION and LMWHA can be confirmed by identifying the 1.6 ppm ^1^H NMR spectra of neat LMWHA (**a**) and LMWHA-SPIONs (**b**). Particle size of neat LMWHA (**c**) is slightly lower than fabricated LMWHA-SPIONs (**d**).

**Figure 3 nanomaterials-12-00496-f003:**
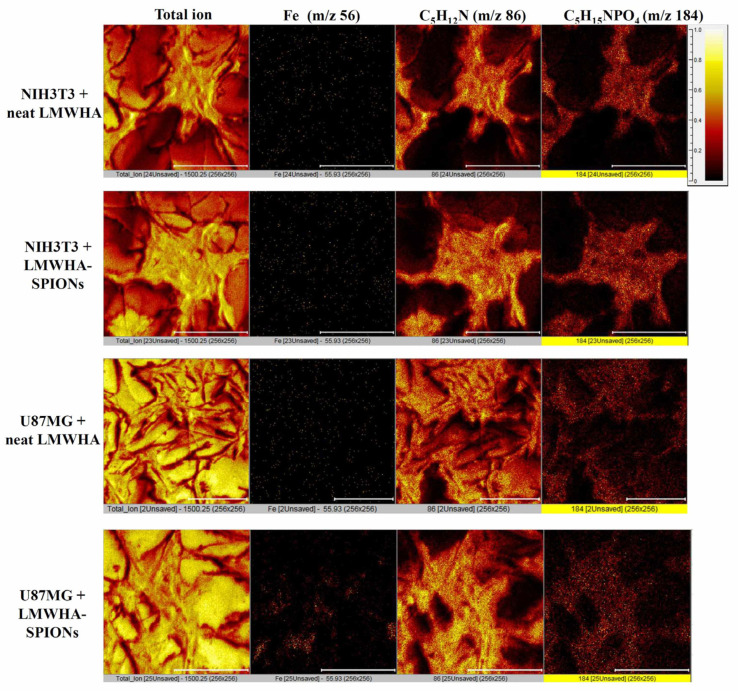
TOF-SIMS images of U87MG and NIH3T3 cells cultured with neat LMWHA and LMWHA-SPIONs. The *m*/*z* 56, 86 and 184 images indicate signals from Fe ions, C_5_H_12_N^+^ and C_5_H_15_NPO_4_^+^ fragments on cellular membranes. (Scale bar indicates 100 μm.).

**Figure 4 nanomaterials-12-00496-f004:**
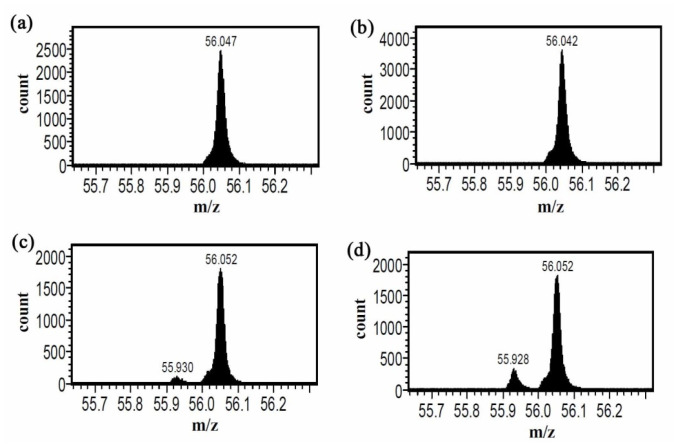
Fe ions can be identified by *m*/*z* 55.93 signals. The background *m*/*z* 56.04 signal comes from the sample holder. When cultured with neat LMWHA, there is no Fe ion signal in both NIH3T3 (**a**) and U87MG cells (**b**). With LMWHA-SPIONs, U87MG sample (**d**) exhibits a higher Fe ion signal compared to NIH3T3 cells (**c**).

**Figure 5 nanomaterials-12-00496-f005:**
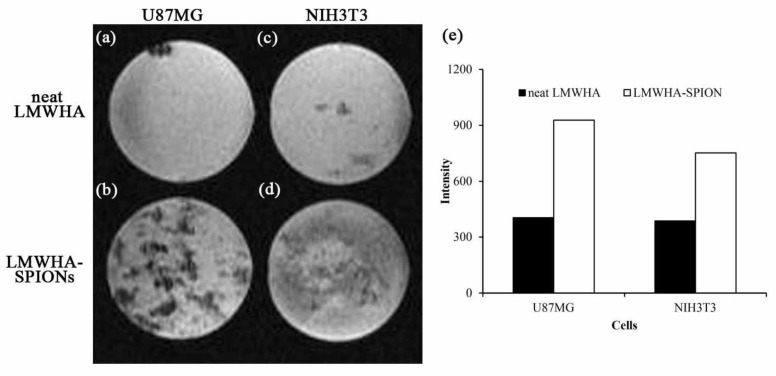
Selective targeting ability of prepared LMWHA-SPIONs on NIH3T3 and U87MG cells can be seen in T2* weighted images. Bright images were obtained when U87MG (**a**) and NIH3T3 (**b**) cells were cultured with neat LMWHA. However, more abundant T2* weighted signals can be observed when LMWHA-SPIONs were used to target the surface of U87MG (**c**) cells cultured with neat LMWHA compared to the NIH3T3 cells (**d**). Quantitative analysis of T2* weighted images showed that the signal intensity of U87MG cells was 1.24 times higher than that of the NIH3T3 cells when the cells were treated with prepared LMWHA-SPIONs (**e**).

**Figure 6 nanomaterials-12-00496-f006:**
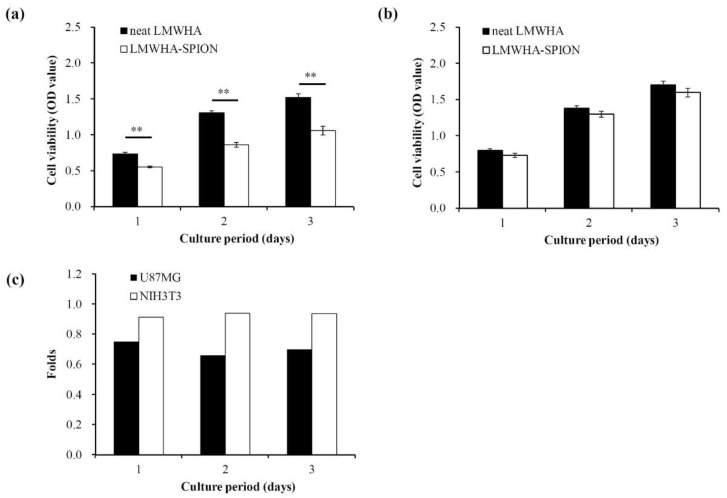
The cell viability of U87MG (**a**) and NIH3T3 (**b**) cells cultured with neat LMWHA and LMWHA-SPIONs. The fold changes in viability when the cells cultured with LMWHA-SPIONs is larger for U87MG cells compared to the NIH3T3 cells (**c**). ** denotes *p* < 0.01.

**Figure 7 nanomaterials-12-00496-f007:**
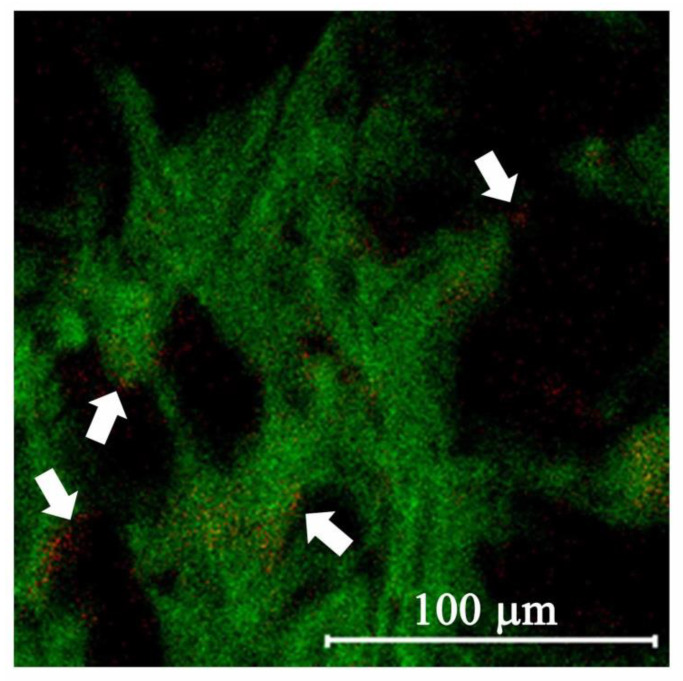
Convolution TOF-SIMS images of Fe ions (*m*/*z* 56 signal with red color) and membrane (*m*/*z* 186 signal with green color) signals indicates that Fe ions aggregate on the surface of U87MG cells. (Scale bar indicates 100 μm.).

## Data Availability

Not applicable.

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
