# Peer review of "Cancer Cytotoxicity of a Hybrid Hyaluronan-Superparamagnetic Iron Oxide Nanoparticle Material: An In-Vitro Evaluation"

_nanomaterials, 2022, doi:10.3390/nano12030496_

Round 1
Reviewer 1 Report
In this work the evaluation of cancer cytotoxicity of a hybrid hyaluronan-superparamagnetic iron oxide nanoparticle material is described. It was found that the LMWHA-SPIONs fabricated in this study specifically bind to U87MG glioblastoma cells and demonstrate an anti-cancer effect on glioblastoma cells. The obtained results in particular LMWHA-SPIONs can be applied as a candidate for both MR imaging applications and as an anticancer agent.
Unfortunately, there are some notes to the article. Authors should avoid any abbreviations in the abstract of the article. The meaning of the abbreviation “CD44 receptor” should be added to the text of the article at the first mention. The scale bar that indicates 100 μm should be presented on the Figure 7. I think that the title of the article “Selective cancer cytotoxicity of a hybrid hyaluronan-superparamagnetic iron oxide nanoparticle material: an in-vitro evaluation” is not correct. What selectivity do the authors mean? I mean, when cytotoxicity is studied in relation to several cancer cells, among which cytotoxicity is observed in relation to one type. In that case the selectivity can be noted. But in this study authors studied cytotoxicity only toward U87MG glioblastoma cancer cells. Therefore I think that the title should be corrected ecpesially the phrase "Selective cancer cytotoxicity ". Summarizing the aforementioned notes, I think that the article looks like a short communication and may be published after minor revision.
Author Response
- Authors should avoid any abbreviations in the abstract of the article. The meaning of the abbreviation “CD44 receptor” should be added to the text of the article at the first mention.
Author response: We thank the favorable comments from the Review. The abstract was rewritten without abbreviation. The full name of CD44 was added on page 1, the last paragraph.
- The scale bar that indicates 100 μm should be presented on the Figure 7.
Author response: The words “100 μm” was added to the scale bar on the Figure 7.
- I think that the title of the article “Selective cancer cytotoxicity of a hybrid hyaluronan-superparamagnetic iron oxide nanoparticle material: an in-vitro evaluation” is not correct. What selectivity do the authors mean? I mean, when cytotoxicity is studied in relation to several cancer cells, among which cytotoxicity is observed in relation to one type. In that case the selectivity can be noted. But in this study authors studied cytotoxicity only toward U87MG glioblastoma cancer cells. Therefore I think that the title should be corrected ecpesially the phrase "Selective cancer cytotoxicity ".
Author response: We thank this comment from the reviewer. In the revised manuscript, the word “selective” has removed from the title and several parts in manuscript.
Reviewer 2 Report
This study is very interesting and brings important contributions to the literature in this field. The characterization techniques have been chosen appropriately with the problem raised. The results are well discussed providing consistent and well-documented explanations. However, to evaluate the chemical structure of the obtained materials FTIR spectra should be presented.
Some other minor corrections:
-the caption of Figure 1e should be corrected: "The absence of hysteresis loop confirms the superparamagnetic behavior of the prepared OA-coated SPIONs at 300 K (e)."
-the authors must use the same abbreviations throughout the manuscript: lines 223, 230, 238, 247, 260, LMWHA-SPIONs should be used.
Author Response
- This study is very interesting and brings important contributions to the literature in this field. The characterization techniques have been chosen appropriately with the problem raised. The results are well discussed providing consistent and well-documented explanations. However, to evaluate the chemical structure of the obtained materials FTIR spectra should be presented.
Author response: We thank the favorable comments from the Review. In this study, NMR was used to evaluate the chemical structure. The NMR spectrum was added on Fig. 2a and b. The Successful binding between oleic acid and LMWHA can be confirmed through the existence of oleic acid long chain proton signals at 1.6 ppm. This descript was presented on page 6, last paragraph.
- The caption of Figure 1e should be corrected: "The absence of hysteresis loop confirms the superparamagnetic behavior of the prepared OA-coated SPIONs at 300 K (e)."
Author response: We thank this comment from the Review. The caption of Figure 1e was revised according to the comment from the reviewer.
- The authors must use the same abbreviations throughout the manuscript: lines 223, 230, 238, 247, 260, LMWHA-SPIONs should be used.
Author response: We thank this comment that point out the typos in our manuscript. All the material name used in the revised manuscript was changed to LMWHA-SPIONs throughout the manuscript.